# The Lactoferrin Phenomenon—A Miracle Molecule

**DOI:** 10.3390/molecules27092941

**Published:** 2022-05-04

**Authors:** Paweł Kowalczyk, Katarzyna Kaczyńska, Patrycja Kleczkowska, Iwona Bukowska-Ośko, Karol Kramkowski, Dorota Sulejczak

**Affiliations:** 1Department of Animal Nutrition, The Kielanowski Institute of Animal Physiology and Nutrition, Polish Academy of Sciences, 05-110 Jabłonna, Poland; p.kowalczyk@ifzz.pl; 2Department of Respiration Physiology, Mossakowski Medical Research Institute, Polish Academy of Sciences, Pawińskiego 5 St., 02-106 Warsaw, Poland; kkaczynska@imdik.pan.pl; 3Maria Sklodowska-Curie, Medical Academy in Warsaw, Solidarności 12 St., 03-411 Warsaw, Poland; 4Military Institute of Hygiene and Epidemiology, 4 Kozielska St., 01-163 Warsaw, Poland; 5Department of Immunopathology of Infectious and Parasitic Diseases, Medical University of Warsaw, Pawinskiego 3c St., 02-106 Warsaw, Poland; ibukowska@wum.edu.pl; 6Department of Physical Chemistry, Medical University of Bialystok, Kilińskiego 1 St., 15-089 Bialystok, Poland; kkramk@wp.pl; 7Department of Experimental Pharmacology, Mossakowski Medical Research Institute, Polish Academy of Sciences, Pawińskiego 5 St., 02-106 Warsaw, Poland

**Keywords:** lactoferrin, oxidative stress, immunomodulation, cancer, aging

## Abstract

Numerous harmful factors that affect the human body from birth to old age cause many disturbances, e.g., in the structure of the genome, inducing cell apoptosis and their degeneration, which leads to the development of many diseases, including cancer. Among the factors leading to pathological processes, microbes, viruses, gene dysregulation and immune system disorders have been described. The function of a protective agent may be played by lactoferrin as a “miracle molecule”, an endogenous protein with a number of favorable antimicrobial, antiviral, antioxidant, immunostimulatory and binding DNA properties. The purpose of this article is to present the broad spectrum of properties and the role that lactoferrin plays in protecting human cells at all stages of life.

## 1. Introduction

Throughout life, from birth to old age, the human body is exposed to a myriad of harmful factors such as toxins, environmental pollutants, pathogens and disease processes including cancer. One common phenomenon that occurs as a result of an imbalance between the production and accumulation of reactive oxygen species (ROS) in cells and tissues and the body’s antioxidant defense mechanisms is oxidative stress. Excessive production and accumulation of ROS is a common denominator of toxicity, being involved in the development and progression of over 100 diseases; therefore, much effort has been devoted to the discovery of effective drugs that prevent the damage induced by ROS [1,2]. These include direct antioxidants or weak oxidants, whose mechanism of action is to increase the activity of endogenous antioxidant mechanisms. Nonetheless, the use of such compounds ends with conflicting epidemiological results in subjects treated with synthetic antioxidants. It is also known that too strong an antioxidant response may not be beneficial [3] as in the case of high-dose vitamin E supplementation, increasing the risk of prostate cancer in healthy men or lung cancer and heart disease in smokers as a result of excessive β-carotene supplementation [4,5].

Therefore, the problem is to maintain a proper balance between ROS and therapeutics with antioxidant properties. Recently, much attention has been focused on lactoferrin (LF), a protein derived from milk, as a potent compound against oxidative stress damage.

Nevertheless, LF is a multifunctional protein that deserves to be called a “miracle molecule”, exhibiting a number of other beneficial properties such as anti-pathogenic, anti-cancer, anti-inflammatory, immunomodulatory and DNA-regulatory activities [6,7]. Recent reports indicate its therapeutic properties in the treatment of neurodegenerative diseases associated with aging, as well as stress-related emotional disorders [8,9].

The purpose of this article was to summarize the wide range of properties of lactoferrin, a “wonder molecule” that plays an important role in health and pathology at all stages of life (Figure 1).

## 2. Lactoferrin—Characteristics and Properties

Lactoferrin (LF) is a component of the whey protein of milk of most mammals, probably with the exception of dogs and rats. The concentration of lactoferrin in milk depends on the phase of lactation. It has been proven that colostrum can contain up to seven times more LF than mature milk [10]. Human body cells can produce lactoferrin and it is also found in many organs and cells of the human body. Its presence has been confirmed in kidneys, lungs, gallbladder, pancreas, intestine, liver, prostate, saliva, tears, sperm, cerebrospinal fluid, urine, bronchial secretions, vaginal discharge, synovial fluid, umbilical cord blood, blood plasma, and cells of the immune system [10,11,12]. It is present wherever the body needs quick and effective protection against external threats.

LF consists of a simple polypeptide chain that consists of two globular lobes with a carboxyl (C) and an amino (N) end connected by an α helix. Each lobe is made of two domains known as C1, C2, N1 and N2, which form a β-sheet [13]. Because of genetic polymorphism and different post-transcriptional and post-translational processing, lactoferrin can occur in many variants [14]. The multifunctionality of LF is due to the fact that it belongs to the class of hybrid proteins, possessing both ordered domains and functionally important intrinsically disordered regions containing sites of various post-translational modifications, such as phosphorylation, acetylation, lipidation, ubiquitination or glycosylation among others, affecting its biological function [15]. An example of LF heterogeneity is the existence of several glycosylation isoforms. The number of N-linked glycosylation sites is species specific. For instance, three potential glycosylation sites have been found in human lactoferrin and five in bovine lactoferrin [13]. Glycosylation plays an important role in regulating LF stability and resistance to proteolysis by increasing the solubility of secreted proteins and increasing the binding of LF to certain cell types or specific receptors. However, it has little effect on properties of LF such as thermal stability or iron binding and release [15].

The major isoform is secreted lactoferrin, which consists of 689 amino acids and has a molecular weight of about 80 kDa. It is odorless and well soluble in water. It shows resistance to heating for several hours at 56 °C, while at temperatures above 80–90 °C it denatures.

LF is a protein with multifaceted effects on the body, not all mechanisms of which have been thoroughly investigated yet, which is why it is referred to as a multipotent protein [16]. Chemically, it is a glycoprotein that, due to its homology of sequence with serum transferrin, is classified as a member of the transferrin family, a protein that can bind to iron ions. Lactoferrin, as the name suggests (lacto + ferrin = milk + iron), is iron binding milk protein, which helps to balance iron levels in the body [11,12,17]. Excess iron can be toxic because it has the ability to donate electrons to oxygen, resulting in the formation of reactive oxygen species (ROS) such as superoxide anions and hydroxyl radicals. LF, thanks to the ability to strongly and reversibly bind iron ions, supports the body in maintaining the homeostasis of this important micronutrient. LF has a high affinity for iron, several hundred times greater than the affinity of transferrin [18], and each lobe of LF can bind to an iron ion [13]. There is a high probability that lactoferrin can also bind copper, zinc and manganese ions [10,19]. In addition, as a result of the chelation process, which reduces the iron overload caused by the accumulation of iron in many organs, leading to free radical generation and dysfunction, the availability of Fe to pathogens that need it for their growth is reduced [20]. Interestingly, LF is capable of reversibly chelating two Fe(III) ions per molecule with high affinity, as well as retaining iron up to pH values as low as 3.0, specific to infectious and inflammatory areas [21].

LF also has the ability to modulate lipid metabolism, which translates not only into better regulation of satiety mechanisms, but also helps fight the tendency to adipose tissue accumulation. The available studies also show a positive role of this protein in reducing fatty liver [22]. It supports the proper functioning of the intestines and increases the absorption of nutrients. Lactoferrin is not only a prebiotic that supports the growth of probiotic bacteria in the digestive system [23,24,25], but also potentiates the effects of some antibiotics (e.g., vancomycin) [26]. Following the antibiotic therapy, LF contributes to the restoration of the balance of the intestinal microbiota and protects against the multiplication of pathogens and the development of local and systemic inflammation [23,24,25].

LF is also known as a factor that promotes osteogenesis and bone health and inhibits the osteolytic process [27,28,29,30].

LF may protect against oxidative stress, which is related to its aforementioned ability to bind iron, which is known to have oxidative properties in large amounts [12,19,31]. The available studies indicate the high effectiveness of lactoferrin in reducing the level of cytotoxins H_2_O_2_ and increasing FRAP (ferric reducing antioxidant power) both in the intra- and extracellular space [32]. It has been shown that another antioxidant mechanism of lactoferrin is its ability to counteract the so-called oxygen explosion in neutrophils, which results in the production of large amounts of free radicals that damage cells [33]. Figure 2 summarizes the beneficial effects of lactoferrin during lifespan.

### 2.1. Immune System—Effects of Lactoferrin on Foetus, Infants and Reproduction

When discussing lactoferrin, it is often said that it is “unable” to affect the body due to digestion; however, scientific evidence suggests that it is hydrolyzed to stable, immunologically active peptides upon contact with the acidic environment of the gastric juice. Lactoferrin preparations are effective after oral administration, which has been confirmed in numerous studies, including clinical ones [32,34,35,36,37,38]. LF can reach the intestine, mainly in the form of peptide fragments, where they act locally on the microbiota and the immune system associated with the local mucosa, thereby enhancing the immunity of all mucous membranes in the body [11,39,40].

In children suffering from diarrhea, oral lactoferrin both alleviated the course and reduced the frequency [40,41]. Published studies using lactoferrin in children have positively evaluated its use for both gastrointestinal infections and sepsis in neonates, and lactoferrin supply in preterm infants is recommended to be introduced as soon as possible [42,43]. It is also very interesting that, in pregnant women, intravaginal lactoferrin is one of the preventive measures reducing the risk of premature delivery [10,41]. LF supports normal tissue development in the fetus, including normal ossification, adequate iron availability and absorption, protects against infection and inflammation, and benefits both mother and fetus [44]. LF acts as a probiotic, protecting the lower genital tract and preventing the consequences of inflammation both during pregnancy and before pregnancy, thus aiding fertility. Lactoferrin is present in the follicular fluid, but results of studies on its effect on oocyte maturation and quality are inconclusive [45,46]. The role of lactoferrin in male fertility is still under intense debate and research [47,48]. It can protect against infections of the male genital tract and regulate iron levels in sperm, thus influencing its quality. This appears to be a good potential marker of sperm quality [47]. This issue is the subject of ongoing clinical trials—ClinicalTrials.gov Identifier: NCT05171504.

There are indications that bovine lactoferrin has the same clinical effects in newborns and infants as human lactoferrin [38]. Fungal infections were also significantly less frequent in children receiving lactoferrin [49].

The two main lines of defense our body relies on when it comes into contact with an antigen are innate and acquired immunity [50]. The former is non-specific, depends on inherited genes and provides some protection from birth. Lactoferrin is an important component of innate immunity [51,52]. Acquired immunity is specific (i.e., dependent on recognized antigens), develops throughout life and allows for a precise response to emerging threats. It is known that the immune system of infants is immature; in contrast to innate immunity, acquired immunity of infants needs to be developed [53]. During the maturation of the immune system, children are supported by their mother’s antibodies. Already in the fetal period, the baby receives a set of IgG immunoglobulins from the mother, which it uses for immunological defense for the first months after birth. However, over time, this supply runs out and the infant’s body has to rely on its own developing immune system. Meanwhile, the baby receives IgA antibodies by consuming their mother’s milk (for this reason they are often referred to as secretory antibodies) [52,54]. As research shows, babies who are breastfed for the first six months are significantly healthier, thanks in part to other substances in breast milk that help support the immune system during this critical period [52,55]. The combination of these three elements; IgG antibodies obtained before birth and IgA contained in breast milk together with lactoferrin constitute a great part of the infant’s immunity in the first months of life [52]. In addition, LF may enhance the effect of antibodies. It has an immunomodulatory potential, both through the influence on the production of cytokines (mainly TNF-alpha, IL-6 and IL-10) and reactive oxygen species, as well as on the functions of lymphocytes and monocytes [56].

LF stimulates the action of receptors *inter alia* for vitamin D, which is very important for osteogenesis and immune response [57]. LF has a protective effect, supports the development of children, especially in the neonatal and infancy period, is a protective factor during gastrointestinal infections and necrotic enteritis in infants [34,38,54,58,59], and reduces the risk of sepsis in infants [35].

For years, clinical trials have been underway based on the use of lactoferrin in the treatment of children, and even extremely premature babies [34,35,36,37,38,49,58,59,60,61]. An important aspect is the lack of reported side effects, which makes lactoferrin a safe protein even for the youngest patients. Administration of bovine lactoferrin to children with low birth weight successfully protects them against sepsis and necrotic enteritis. The properties of lactoferrin make it not only an important factor regulating the work of our body, but also a potential therapeutic tool [38].

It seems necessary to re-emphasize that studies with the use of lactoferrin mainly include tissue models and animal models, as well as clinical trials involving newborns, infants and children. But we are still unsure of its potential for adults. The supplementation of that group of human population still needs further detailed studies.

### 2.2. Antitoxic and Antipathogenic Properties

Numerous studies have confirmed the beneficial effects of LF on the intestinal epithelium. This protein stimulates the growth, differentiation and secretory activity of epithelial cells, which optimizes the digestive processes and absorption of nutrients and protects against the action of pathogens and food allergens [61,62]. LF also protects the intestinal epithelium from the toxic effects of reactive oxygen species (ROS), bacterial toxins and xenobiotics such as nonsteroidal anti-inflammatory drugs (NSAIDs) [31,63,64,65]. Importantly, LF also protects against gastrointestinal tract infections, both viral and bacterial, fungal and protozoal [32,66]. Many tests have demonstrated the protective effect of LF in the states of endotoxemia, bacteremia, sepsis and necrotic enteritis in neonates [35,36,58,67,68,69], in inflammatory colitis [70,71] and after partial bowel resection [72]. LF has antibacterial properties in relation to Gram-negative and Gram-positive bacteria, thanks to which it is helpful for fighting pathogens, prevents the formation of biofilm by pathogenic bacteria, such as *Staphylococcus aureus* or blue oil rod (*Pseudomonas aeruginosa*) [66,73]. LF supports the treatment of gastric infection caused by *Helicobacter pylori* [74]. The mechanism of action of LF may, *inter alia*, include the direct inhibition or killing of microbial cells, activation/inhibition of the immune system, or enhancement of intestinal epithelial tightness by stimulating the production of tight junction proteins. In addition, the binding of iron by LF makes its absence associated with a concomitant halt in bacterial growth, which protects the body from infection. [20]. It also has an immunomodulating effect, stimulating the body to synthesize cytokines and chemokines as well as accelerating the maturation of cells of the immune system [17,50,51,75].

Human lactoferrin (abbreviated as hLF,) possesses 77% similarities with the bovine form (bLF) in the aspect of amino acid sequences, although bovine lactoferrin is usually studied, because it is easier to obtain. It has been estimated that in a glass of cow’s milk we will find about 25-75 mg of this protein. At the same time, it seems that bLF is not an ideal choice due to differences that may alter its antiviral and antimicrobial potential when used in human therapy, but some authors highlight its stronger antimicrobial activity [76].

LF also has antiviral properties and works synergistically with antiviral drugs such as acyclovir, ribavirin or zidovudine [77]. LF is able to bind to receptors, such as ACEII (Figure 1), used by SARS-CoV-2 as an anchor site in the cell membrane and thus inhibit the adsorption of the pathogen to the cell. In addition, LF is able to block the pathogen’s surface receptors and prevent it from binding to the target cell [78,79,80].

Its antifungal properties, including against dermatophytes and supporting the action of antimycotic drugs has been shown [49,81]. In addition it has an antiparasitic effect, e.g., against the motile parasite (*Plasmodium vivax*) that causes malaria and the protozoan *Toxoplasma gondii* causing toxoplasmosis [82].

For more detailed information on the LF activity against bacteria, viruses, fungi, and parasites, see the review by Gruden and Urlih [83].

### 2.3. Anticancer Activity

One of the many properties of LF is its anticancer activity. This may be related not only to preventing antioxidant stress and inflammation, which contribute to DNA damage and tumorigenesis, but also to preventing the development of, or inhibiting, cancer by stimulating the adaptive immune response [7]. This is the case for colorectal cancer, the epidemiology of which is mainly related to age and lifestyle factors [84] and in the case of childhood leukemia where long-term consumption of breast milk may prevent the risk of developing leukemia due to the immunoprotective properties of the LF present [85]. Furthermore, LF may directly inhibit proliferation, survival, migration, metastasis and accelerating cancer cell death [86,87].

It has been confirmed that, in the presence of LF, various cancer cells undergo remarkable damage such as cell cycle arrest, damage to the cytoskeleton and induction of apoptosis, as well as decreased cell migration [13,86]. The postulated property of LF by which it activates signaling pathways to generate deleterious effects on cancer cells may be interaction with proteoglycans, glycosaminoglycans, and sialic acid, high levels of which are presented by cancer cells. This may also explain the high cytotoxic selectivity of LF against cancer cells only [13,87,88]. Besides, the ability of LF to enter the cell nucleus is likely the primary mechanism by which it exerts its pleiotropic functions, including anticancer. Nuclear LF (called delta) acts as a transcription factor and causes activation of expression target genes such as Bax, SelH, DcpS, UBE2E1, Skp1 and GTF2F2 and shows the anticancer, anti-proliferating and pro-apoptotic activities [89,90,91,92]. This corresponds to decreased levels of LF and delta LF expression in tumor cells, which often correlates with greater tumor progression and poor prognosis [91,93,94].

LF also binds iron, which is heavily involved in the metabolic requirements of some cancer cells, and blocks angiogenesis, i.e., prevents the formation of new blood vessels, thereby inhibiting tumor growth and metastasis or directing the tumor toward apoptosis [21,95].

An interesting feature of LF that deserves attention, in addition to its proven safety and its low antigenicity and selectivity for cancer cells, which could be used in brain tumor therapy, is its passage through the blood–brain barrier [87].

### 2.4. Aging and Aging-Related Diseases

Aging can be defined as: “the progressive accumulation of changes with time associated with or responsible for the ever-increasing susceptibility to disease and death which accompanies advancing age”, and the factors that lead to aging: “the sum of the deleterious free radical reactions going on continuously throughout the cells and tissues constitutes the aging process or is a major contributor to it” [96] and “changes in molecular structure and, hence, function” [97]. In summary, aging is a complex natural phenomenon occurring as a consequence of the passage of time, environmental factors and genetics that increase susceptibility to developing systemic diseases, including metabolic disorders (diabetes mellitus), cardiovascular, neurodegenerative and respiratory diseases as well as rheumatoid arthritis, cancers or dementia [8,98,99,100].

The pleiotropic anti-aging effect of LF is related to its antioxidant, anti-inflammatory and anticancer effects, as well as the assurance of neuroprotection or the alleviation of mitochondrial dysfunction and systemic disorders [8].

LF antioxidant potential leads to cells’ and organs’ protection finally extending its lifespan [101]. In addition, due to regulation of numerous genes expression (inhibition of NF-κB, mTORC1 and caspase via the Erk and Akt pathways), LF regulates cell growth, proliferation, apoptosis and inflammation. It suppresses the senescence and apoptosis of mesenchymal stem cells (MSCs) [102,103], promotes both the formation of granulation tissue and re-epithelialization (proliferation and migration of fibroblasts and keratinocytes stimulation and enhancement of extracellular matrix components synthesis) [104,105]. Moreover, due to the induction of the targeted apoptosis of senescent cells or the disruption of the senescence-associated secretory phenotype (SASP), LF restores tissue homeostasis [99,100]. Interestingly, LF usefulness has been shown for treatment, diagnosis or monitoring age-related diseases [106,107,108,109,110,111,112,113,114,115,116,117,118,119,120]. For example, it may act as a neuroprotective agent in Alzheimer’s disease (AD) and Parkinson’s disease (PD) [111,115,116] leading to the improvement of cognitive function and attenuation of brain senescence [121]. The possible mechanism of LF action includes iron-binding dependent manner (upregulation of divalent metal transporter 1 (DMT1) and transferrin receptor (TFR) and downregulation ferroportin 1 (Fpn1)) [117,118] and/or iron-binding independent manner (regulation of the p-Akt/PTEN or the ERK-CREB pathway in HIF-1-dependent manner) [118,119,120]. Furthermore, LF preserves mitochondrial calcium homeostasis in degenerated dopaminergic neurons [122]. Moreover, LF regulates body fat metabolism limiting obesity (probable downregulation of adipogenic genes and upregulation of fatty acid synthase and acetyl CoA carboxylase in adipocytes) [123,124] and glucose metabolism in patients with type 2 diabetes mellitus via improvement of the insulin-signaling response in adipocytes (up-phosphorylation of Akt serine 473 and up-expression of glucose transport 4 and insulin receptor 1) [107,124,125,126]. In patients with cardiovascular diseases, bLF exerts proangiogenic effects and reduces blood pressure [127]. Interestingly, LF delays the process of senile osteoporosis due to its antioxidant effect and inhibition of osteoblast senescence related genes (IGF1 signaling pathway) [128,129]. LF function as a biomarker molecule is presented in Table 1.

### 2.5. Lactoferrin in the Human Diet and Therapy of Diseases

Lactoferrin is an important component in the human diet. Due to its high nutritional value, its antibacterial, antiviral, anti-cancer properties and regulation of the activity of the immune system [17], it has also been used in the pharmaceutical and food industries and in the production of feed additives. Currently, we can find it in products such as dietary supplements and infant formula. Lactoferrin obtained from cow’s milk is used, among others, in the production of infant formulas, foodstuffs for special medical purposes, milk, yoghurt drinks, ice cream and cookies, dietary supplements, and processed cereal products. It is also appreciated in the cosmetic (e.g., in cosmetics and toothpaste) and pharmaceutical industries [42,44,130,131].

It is a safe raw material, which is confirmed by the documents issued by the European Parliament (EP) [132], European Food Safety Authority (EFSA) [133] and the Food and Drug Administration (FDA) in the United States [134]. However, bovine LF, like any protein in cow’s milk, can cause an allergy, which is often called protein blemish. Therefore, the consumption of lactoferrin carries the risk of abnormal reactions in the body, and preparations containing this ingredient should not be taken in the case of suspected or diagnosed allergy to cow’s milk proteins. On the other hand, people who do not have allergies, but suffer from lactose intolerance, can choose preparations with LF, with a clearly defined lack of milk sugar in its composition [26,135,136,137,138].

It is also worth mentioning that LF, as a naturally occurring protein in saliva and produced by salivary glands in the oral cavity, has protective properties and is supposed to provide homeostasis in the oral cavity [17]. The ability to bind iron ions by LF provides antibacterial activity in the oral cavity. The use of products with LF further supplements it in the oral or nasal cavity, thus strengthening the first protective barrier against bacteria and viruses from the outside [60,139,140,141,142,143]. The effectiveness of LF has been established in numerous in vitro, animal, and human studies in which LF, used in oral and vaginal formulations, positively altered the ecosystem of the reproductive tract by eliminating pathogenic microorganisms and increasing Lactobacillus species, re-establishing the state of eubiosis and protecting from dangerous consequences of dysbiosis, such as premature labor or miscarriage [6,144].

So far, most of the data on the positive effects of LF in pathological conditions are mainly based on studies in animal models. To date, studies in animal models have shown a significant increase in survival in rodents when sepsis developed after injection of *E. coli* [145]. Subsequent work revealed the strong anti-inflammatory effect of LF in models with induced gastritis or enteritis. However, the most promising results come from experiments based on the administration of lactoferrin to subjects with immature digestive systems (possibly due to an underdeveloped intestinal microbiome). Calves and newborn rats were characterized by better absorption of nutrients and a significant increase in intestinal villi length and stimulation of the development of the immune system [40,142,146,147,148,149]. Lactoferrin was reported to protect against oxidative stress-induced mitochondrial dysfunction and DNA damage, thus modulating innate immune responsiveness which can further alter the production of immune regulatory mediators that are important for directing the development of adaptive immune function [12,19,31,39,75,150]. Such LF action was revealed both in cell culture and within an animal model of endotoxemia. In fact, mitochondria from lipopolysaccharide (LPS)-treated animals released significantly higher amounts of H_2_O_2_ than those isolated from LF-pre-treated plus LPS-challenged animals [150]. This mechanism is of fundamental protective importance at the beginning of an infection. After the infection phase, lactoferrin shows a strong immunotropic effect: it stimulates the cells of the immune system to mature rapidly and enhance the immune response.

In order to not rely only on animal studies, it is worth recalling clinical trials. Among adults, it was possible to notice an improvement in the condition of people suffering from chronic *H. pylori* infection (the most common cause of peptic ulcer disease) in a form resistant to conventional treatment [151]. The results of studies involving patients suffering from various types of cancer are also promising, although preliminary. The anticancer effects of supplementation with LF in the gastrointestinal tract cancer and protection against colon cancer, stomach cancer, liver cancer and pancreatic cancer may be explained by the antioxidant properties of lactoferrin [12,152,153,154,155] (Figure 3).

Nevertheless, if we consider using LF in therapy, the form of its administration is very important. Lactoferrin is a hydrophilic substance, therefore in the non-liposomal version it has very limited absorption from the stomach [19]. In free form, it is decomposed there by hydrochloric acid and enzymes (proteases). Therefore, the bioavailability of the free form of lactoferrin may be limited. The use of small liposome vesicles may be beneficial in this case [156,157,158]. Nanoliposomes protect lactoferrin from destruction by digestive juices, allowing the intact protein to pass into the duodenum, from there into the general circulation, ensuring its high bioavailability [159] and impact on iron ions homeostasis, the skeletal system and, of course, the immune system.

LF administered in a phosphatidylcholine encapsulated form also has the potential to penetrate deep into the mucosa, and due to the small size of the nanoliposome (100 nm) compared to the virus size (150 nm), it is more competitive in reaching receptors on target cells where it settles in front of the virus [156,157]. It is important to note that the mucous membranes lining the oral or nasal cavities are very permeable, so this additional protection against viruses based on nanolactoferrin is very relevant.

## 3. Conclusions

Lactoferrin is a multifunctional protein derived from milk with high affinity for iron ions. It is known that iron is necessary for microorganisms to grow and reproduce, so the sequestration of iron significantly reduces their pathogenic potential. LF has numerous beneficial properties—antibacterial, antiviral, antifungal and antiparasitic, as well as immunomodulatory, anti-inflammatory and anticancer properties—that may play an important role in maintaining health from fetal life to old age. Currently, LF is an ingredient in many supplements and medicines, but a thorough understanding of the mechanisms of its beneficial effects requires further in-depth research.

## Figures and Tables

**Figure 1 molecules-27-02941-f001:**
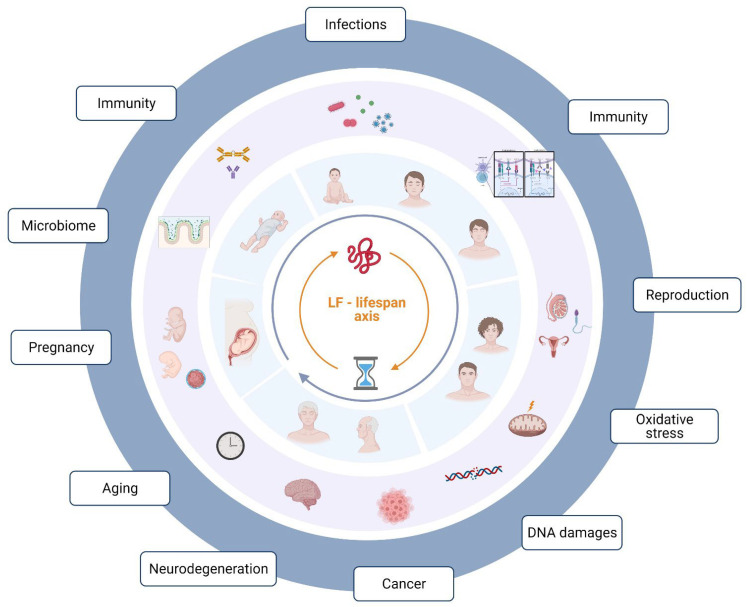
Lactoferrin in human physiological states and pathology—lifespan correlation. Illustration was created in BioRender.com (accessed on 28 April 2022).

**Figure 2 molecules-27-02941-f002:**
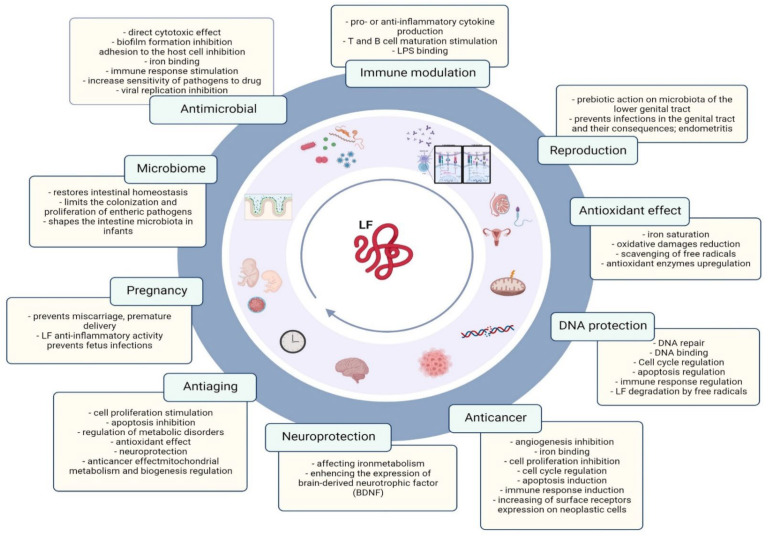
Lactoferrin properties related to the human lifespan. Illustration was created in BioRender.com.

**Figure 3 molecules-27-02941-f003:**
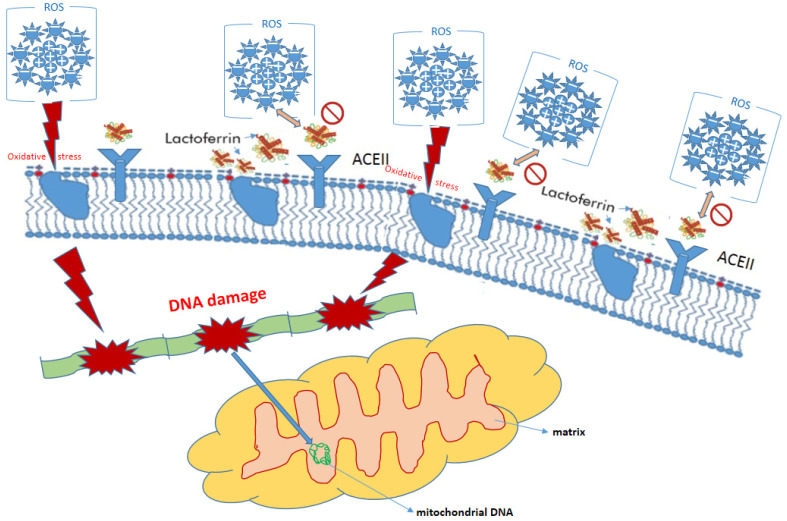
Protective role of lactoferrin in eukaryotic cell.

**Table 1 molecules-27-02941-t001:** Increased level of LF in the diagnosis and monitoring of aging-associated diseases.

LF Localisation	Disorders	Disease	Corellation	Refs.
plasma	metabolic	type 2 diabetes mellitus	insulin sensitivity (+)	[106,107]
			ischemic heart disease	[108]
			plasma glucose (−)	
		Metabolic	fasting triglycerides, glucose, and body composition (-)	[106]
			high density lipoprotein cholesterol (+)	
plasma	cardiovascular		lipemia (+)	[109]
			ischemic stroke (+)	[110]
			the risk for cardiovascular events (+)	
saliva	neurodegenerative	Alzheimer’s disease	cognitive impairment (+)	[111]
brain			Diagnosis (+)	
cerebrospinal fluid		Parkinson’s disease	Diagnosis (+)	[112]
faecal	inflammatory	Crohn’s disease	Diagnosis (+)	[113]
synovial		rheumatoid arthritis	Diagnosis (+)	[114]

(+) positive correlation; (−) negative correlation.

## Data Availability

On request of those interested.

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
