# Peer review of "The Lactoferrin Phenomenon—A Miracle Molecule"

_molecules, 2022, doi:10.3390/molecules27092941_

Round 1

Reviewer 1 Report

I find the review very interesting as it describes several of the most important functions of lactoferrin. However, it would be appropriate to discuss the structure of the molecule in greater depth, especially its glycosylated structure, since several of the functions described here may be related to this glycosylated structure. So I recommend that at least two or three lines be written to describing this part. You can consult the following review:

Rascón-Cruz, Q., Espinoza-Sánchez, E. A., Siqueiros-Cendón, T. S., Nakamura-Bencomo, S. I., Arévalo-Gallegos, S., & Iglesias-Figueroa, B. F. (2021). Lactoferrin: A glycoprotein involved in immunomodulation, anticancer, and antimicrobial processes. Molecules, 26(1), 205.

Author Response

Reviewer 1

I find the review very interesting as it describes several of the most important functions of lactoferrin. However, it would be appropriate to discuss the structure of the molecule in greater depth, especially its glycosylated structure, since several of the functions described here may be related to this glycosylated structure. So I recommend that at least two or three lines be written to describing this part. You can consult the following review:

Rascón-Cruz, Q., Espinoza-Sánchez, E. A., Siqueiros-Cendón, T. S., Nakamura-Bencomo, S. I., Arévalo-Gallegos, S., & Iglesias-Figueroa, B. F. (2021). Lactoferrin: A glycoprotein involved in immunomodulation, anticancer, and antimicrobial processes. Molecules26(1), 205.

-Dear Reviewer, Thank you for your positive review. We found the above paper interesting and have cited it in several places in the text. We have also included a description of the LF molecule and its glycosylated structure (lines 82-94).

Reviewer 2 Report

The title of this manuscript is high-sounding while the content is poor, not new (there are too many reviews on lactoferrin) and the bibliography is old, incomplete and, sometimes, misinterpreted.

The Authors also ignore very important recent publications.

The manuscript cannot be published.

Author Response

Reviewer 2

The title of this manuscript is high-sounding while the content is poor, not new (there are too many reviews on lactoferrin) and the bibliography is old, incomplete and, sometimes, misinterpreted. The Authors also ignore very important recent publications.

The manuscript cannot be published.

-Dear Reviewer thank you for your comments. The paper has bee substantially rewritten. The idea was to show the most important beneficial effect of lactoferrin for human body from birth (even from fetal life) to old age. More recent literature has been also added. We hope the article is more satisfactory in its current form.

Reviewer 3 Report

This comprehensive review describes multiple functions of lactoferrin, which the authors address as a "miracle molecule". Although the manuscript contains a good amount of useful information, there are several issues that need to be addressed.

1) Structural heterogeneity and multifunctionality of lactoferrin was carefully analyzed in a comprehensive review published in 2014 (PMID: 25245670), where in addition to the description of the peculiar structural features of this protein, the roles intrinsically disordered regions were emphasized. This prior work is not cited in the manuscript. In my view, at least some brief discussion of this review should be included, as it contains some mechanistic explanations of the multifunctionality of lactoferrin and some of the related arguments can be used to support the miracle molecule view of this protein.

2) Manuscript has multiple linguistic issues and requires careful editing.

3) In my view, some more illustrative material should be added. 

Author Response

Reviewer 3

This comprehensive review describes multiple functions of lactoferrin, which the authors address as a "miracle molecule". Although the manuscript contains a good amount of useful information, there are several issues that need to be addressed.

1)Structural heterogeneity and multifunctionality of lactoferrin was carefully analyzed in a comprehensive review published in 2014 (PMID: 25245670), where in addition to the description of the peculiar structural features of this protein, the roles intrinsically disordered regions were emphasized. This prior work is not cited in the manuscript. In my view, at least some brief discussion of this review should be included, as it contains some mechanistic explanations of the multifunctionality of lactoferrin and some of the related arguments can be used to support the miracle molecule view of this protein.

- Dear Reviewer thank you for your suggestions. This article has been added and cited, and discussion of the peculiar structural features of LF protein has been included (lines 82-94).

2) Manuscript has multiple linguistic issues and requires careful editing.

- The language of the article has been revised.

3) In my view, some more illustrative material should be added. 

- More illustrations have been added.

Round 2

Reviewer 2 Report

Dear Authors,

The paper has been improved and after the addition of these references, the manuscript can be accepted.

Concerning the in vitro experiments on lactoferrin against SARS-CoV-2, the following references must be cited: 

  • Mirabelli et al. 2021 Morphological cell profiling of SARS-CoV-2 infection identifies drug repurposing candidates for COVID-19. Proc Natl Acad Sci U S A 118(36):e2105815118. doi: 10.1073/pnas.2105815118
  • Campione et al. 2021 Lactoferrin Against SARS-CoV-2: In Vitro and In Silico Evidences. Front Pharmacol 12:666600. doi:10.3389/fphar.2021.666600

Moreover, the in vivo trials must be cited:

  • Algahtani et al 2021 The Prospect of Lactoferrin Use as Adjunctive Agent in Management of SARS-CoV-2 Patients: A Randomized Pilot Study. Medicina (Kaunas) 57(8):842. doi:10.3390/medicina57080842
  • Campione et al. 2021 Lactoferrin as Antiviral Treatment in COVID-19 Management: Preliminary Evidence. Int J Environ Res Public Health 18(20):10985. doi:10.3390/ijerph182010985
  • Rosa et al. 2021 Ambulatory COVID-19 Patients Treated with Lactoferrin as a Supplementary Antiviral Agent: A Preliminary Study. J Clin Med 10(18):4276. doi: 10.3390/jcm10184276

Reviewer 3 Report

All concerns were adequately addressed, new figures were added and the manuscript was amended accordingly.